# QTL Mapping for Seedling and Adult Plant Resistance to Leaf and Stem Rusts in Pamyati Azieva × Paragon Mapping Population of Bread Wheat

**Yuliya Genievskaya** [1], **Saule Abugalieva** [1,2], **Aralbek Rsaliyev** [3], **Gulbahar Yskakova** [3] **and Yerlan Turuspekov** [1,4,*]

[1] Laboratory of Molecular Genetics, Institute of Plant Biology and Biotechnology, Almaty 050040, Kazakhstan; julia.genievskaya@gmail.com (Y.G.); absaule@yahoo.com (S.A.)
[2] Department of Biodiversity and Bioresources, Faculty of Biology and Biotechnology, al-Farabi Kazakh National University, Almaty 050040, Kazakhstan
[3] Laboratory of Phytosanitary Safety, Research Institute of Biological Safety Problems, Gvardeisky 080409, Zhambyl Region, Kazakhstan; aralbek@mail.ru (A.R.); y_gulbahar@mail.ru (G.Y.)
[4] Agrobiology Faculty, Kazakh National Agrarian University, Almaty 050010, Kazakhstan
[*] Correspondence: yerlant@yahoo.com; Tel.: +7-727-394-8006

**Abstract:** Leaf rust (LR) and stem rust (SR) pose serious challenges to wheat production in Kazakhstan. In recent years, the susceptibility of local wheat cultivars has substantially decreased grain yield and quality. Therefore, local breeding projects must be adjusted toward the improvement of LR and SR disease resistances, including genetic approaches. In this study, a spring wheat segregating population of Pamyati Azieva (PA) × Paragon (Par), consisting of 98 recombinant inbred lines (RILs), was analyzed for the resistance to LR and SR at the seedling and adult plant-growth stages. In total, 24 quantitative trait loci (QTLs) for resistance to rust diseases at the seedling and adult plant stages were identified, including 11 QTLs for LR and 13 QTLs for SR resistances. Fourteen QTLs were in similar locations to QTLs and major genes detected in previous linkage mapping and genome-wide association studies. The remaining 10 QTLs are potentially new genetic factors for LR and SR resistance in wheat. Overall, the QTLs revealed in this study may play an important role in the improvement of wheat resistance to LR and SR per the marker-assisted selection approach.

**Keywords:** *Triticum aestivum*; QTL; mapping population; leaf rust; stem rust; pathogen races; disease resistance

## 1. Introduction

Bread wheat (*Triticum aestivum* L.) is one of the major cereal crops in the world. In 2018/2019, the global production of wheat was 734.7 million metric tons, ranking second place amongst the grains after maize [1]. It is used mostly as flour for the production of a large variety of leavened and flat breads and the manufacturing of a wide range of other baked products [2]. In 2018/2019, Kazakhstan was ranked the 12th largest wheat producer in the world [3]. In Kazakhstan, wheat is cultivated on about 13 million hectares annually. The country produces up to 20–25 million tons of bread wheat per year and exports up to 5–7 million tons of the grain [4]. The primary goals of modern wheat breeding programs worldwide include enhancing grain yield and quality and increasing resistance to biotic and abiotic stresses to ensure global food security [5]. Biotic stresses include dangerous fungal diseases and particularly the most common representatives of the *Puccinia* genus: *Puccinia recondita* Rob. ex Desm f. sp. *tritici*, causing leaf rust (LR), and *Puccinia graminis* Pers. f. sp. *tritici* Eriks. & Henn., which is responsible for stem rust (SR) of wheat.

LR generally causes light to moderate yield losses ranging from 1% to 20% over a large area, but when the disease is severe prior to heading time, it may destroy up to 90% of the wheat crop [6]. For example, in Kazakhstan, epiphytotic development of the pathogen on spring wheat during 2000–2001 resulted in 50–100% LR severity on commercial cultivars in the Akmola region (northern Kazakhstan), which is the main wheat-growing region in the country [7].

SR is another important rust disease that is often considered the most devastating of the wheat rust diseases because it can cause complete crop loss over a large area within a short period of time [8]. In 2016, northern Kazakhstan was subjected to an epiphytotic outbreak of SR, resulting in 50% disease development severity in the field, decreasing wheat yield and grain quality [7]. Nowadays, local farmers prefer the usage of fungicides to protect wheat fields from LR and SR; however, this method is harmful to the environment and more expensive than breeding and growing genetically resistant wheat cultivars [9].

LR and SR resistances are controlled by a diverse group of genes, designated as *Lr* and *Sr*, respectively [10]. In the last 100 years, approximately 80 *Lr* resistance genes have been identified and described in bread wheat, durum wheat, and diploid wheat species [10], and the list is still growing. For SR, nearly 60 *Sr* genes have been identified to date in wheat and its wild relatives [10]. Generally, resistance to rust diseases can be broadly categorized into two types. The first is resistance at all growth stages (called seedling resistance), detected at the seedling stage and expressed until the plant dies. This type of resistance is controlled by the R type of genes, and the majority of *Lr* and *Sr* genes belong to this group. The efficacy of the R gene is pathogen-strain-dependent [11]. The second type of resistance is adult plant resistance (APR), where genes are ineffective during the seedling stage but provide robust resistance at maturity [11]. For example, LR resistance genes *Lr12*, *Lr13*, *Lr22a*, *Lr22b*, *Lr34*, *Lr35*, *Lr46*, *Lr48*, *Lr49*, and *Lr67*, and SR resistance genes *Sr2* and *Sr57* are well-characterized APR genes [10]. Durable rust resistance is more likely to be the APR type rather than the seedling type [12]; both types are important for wheat breeding [11].

Two of the most effective methods of quantitative trait locus (QTL) mapping are based on association panels and biparental segregating populations [13]. Both of these methods provide the means to investigate the genome and describe the etiology of complex quantitative traits, including disease resistance [14–17]. Genetic maps are a key tool enabling genetic linkage studies and searches for novel loci responsible for traits. Modern high-throughput sequencing technologies allow for the high-accuracy genotyping of large collections with genetically diverse germplasms [18,19] and segregating mapping populations, such as doubled haploids (DHs), recombinant inbred lines (RILs), $F_2$, and backcross (BC) populations [20]. Linkage maps were successfully used for the QTL analyses of wheat yield components [21], grain quality traits [22], abiotic [23], and biotic stress factors, including pests [24].

The primary goal of this study was to identify QTLs involved in seedling and adult plant resistance of bread wheat to LR and SR under environmental conditions in southern and southeastern Kazakhstan. To meet this goal, the Pamyati Azieva × Paragon (PA × Par) RILs mapping population (MP) was studied in field and greenhouse (GH) conditions. Previously, this population was successfully used for the analysis of yield-related traits [25] and adult plant resistance to LR and SR in south-east and northern Kazakhstan in 2018 [26,27]. Hence, the current study adds the investigation of seedling resistance in the MP to LR and SR races. In addition to one-year studies in south-east and north Kazakhstan, this work covers the analysis of LR and SR resistance in the MP in southern Kazakhstan in 2018 and 2019.

## 2. Materials and Methods

### 2.1. Plant Material and Genotyping

The biparental mapping population PA × Par composed of 98 RILs was developed in greenhouse conditions of the John Innes Centre (Norwich, UK) during 2011–2015 under the ADAPTAWHEAT

project [28]. The RIL population was obtained via a single-seed descent method using two parental cultivars: Paragon (elite spring cultivar originated from the U.K.) and Pamyati Azieva (a commercial spring cultivar originating from Russia and registered in Kazakhstan) [29]. Both cultivars were chosen due to their diverse genetic backgrounds and different manifestations of yield traits as well as resistance to diseases. The RIL population was further developed for $F_8$ generation in the fields of southeastern Kazakhstan.

The RILs and two parental cultivars were genotyped using the Illumina's iSelect 20K single nucleotide polymorphism (SNP) array at the TraitGenetics Company (TraitGenetics GmbH, Gatersleben, Germany). The genotypic data were filtered from markers with >10% missing data and with <0.1 minor allele frequency and consisted of 4595 polymorphic SNP markers.

## 2.2. Phenotyping of Seedling Resistance in Greenhouse

For the comprehensive study of the PA × Par MP response to LR and SR pathogens, the resistance was evaluated at the seedling and adult plant-growth stages. Race-specific resistance at the seedling stage was assessed in a greenhouse of the Research Institute of Biological Safety Problems (RIBSP, Gvardeisky, Zhambyl region, southern Kazakhstan). For the inoculation of RILs seedlings (7–10 days after sowing) in greenhouse conditions, three races of *P. graminis* and three races of *P. recondita* with different levels of virulence to *Sr* and *Lr* genes, respectively, were used (Table 1). Inoculated plants were placed in the boxes of the greenhouse with appropriate temperature conditions (22 ± 2 °C for SR, 18 ± 2 °C for LR) and illumination (10,000–15,000 lux, 16 h' light period) [30–32]. RIL reaction was assessed on the 14th day after inoculation, according to the scale reported by Stakman [33]. The experiment was performed in two independent replicates.

**Table 1.** Virulence/avirulence pattern of pathogen races used in the study.

| Disease (Pathogen) | Race | Avirulent (Effective) Genes | Virulent (Ineffective) Genes |
|---|---|---|---|
| LR (*Puccinia recondita* Rob. ex Desm f. sp. *tritici*) | TQKHT | *Lr24, 26, 3ka, 19, 25* | *Lr1, 2a, 2c, 3, 9, 16, 11, 17, 30, 20, 29, 2b, 3bg, 14a, 15* |
| | TRTHT | *Lr24, 19, 25* | *Lr1, 2a, 2c, 3, 9, 16, 26, 3ka, 11, 17, 30, 20, 29, 2b, 3bg, 14a, 15* |
| | TQTMQ | *Lr24, 26, 20, 25, 14a, 15* | *Lr1, 2a, 2c, 3, 9, 16, 3ka, 11, 17, 30, 19, 29, 2b, 3bg* |
| SR (*Puccinia graminis* Pers. f. sp. *tritici* Eriks. & E. Henn.) | TKRTF | *Sr11, 30, 24, 31* | *Sr5, 21, 9e, 7b, 6, 8a, 9g, 36, 9b, 17, 9a, 9d, 10, 38, Tmp, McN* |
| | PKCTC | *Sr21, 11, 36, 9b, 30, 24, 31, 38* | *Sr5, 9e, 7b, 6, 8a, 9g, 17, 9a, 9d, 10, Tmp, McN* |
| | RKRTF | *Sr9e, 11, 30, 24, 31* | *Sr5, 21, 7b, 6, 8a, 9g, 36, 9b, 17, 9a, 9d, 10, 38, Tmp, McN* |

LR, leaf rust; SR, stem rust.

Races of *P. graminis* were differentiated in 2018 [34] using the North American nomenclature [35] with the assistance of five sets of SR-differentiating wheat cultivars. Races of *P. recondita* were also identified in 2018 using 20 Thatcher near-isogenic lines (NILs) sets of *Lr* genes [36–38]. For the nomenclature of *P. recondita* races, Virulence Analysis Tools [39] were used.

## 2.3. Adult Plant Resistance and Yield Components in Field Conditions

APR in the field was tested in two environments: the RIBSP and the Kazakh Research Institute of Agriculture and Plant Industry (KRIAPI, Almalybak, Almaty region, southeastern Kazakhstan) (Table 2).

**Table 2.** Meteorological data on average temperature and precipitations during the vegetation period in the fields of Research Institute of Biological Safety Problems (RIBSP, southern Kazakhstan) and Kazakh Research Institute of Agriculture and Plant Industry (KRIAPI, southeastern Kazakhstan).

| | KRIAPI | | | | |
|---|---|---|---|---|---|
| | **March** | **April** | **May** | **June** | **July** |
| Temperature (°C) | 8.2 | 12.8 | 17.0 | 22.3 | 27.0 |
| Precipitation (mm) | 27.3 | 168.4 | 39.3 | 72.7 | 22.6 |
| | RIBSP | | | | |
| | **March** | **April** | **May** | **June** | **July** |
| Temperature (°C) | 13.0 | 17.0 | 24.0 | 29.0 | 33.0 |
| Precipitation (mm) | 1.0 | 2.3 | 1.8 | 0.9 | 0.1 |

In RIBSP fields, mixed races of LR and SR urediniospores common in Kazakhstan were applied as inoculum. The inoculum was activated at a temperature of 37–40 °C for 30 min, followed by watering in a humid chamber at a temperature of 18–22 °C for 2 h. At the booting stage, individual plants were treated with an aqueous suspension of leaf and stem rust urediniospores dissolved in Tween 80 detergent. After inoculation, the plots were covered with plastic wrap for 16–18 h. In KRIAPI fields, inoculation occurred with local LR and SR pathogen populations in uncontrolled natural conditions.

Thus, experiments were conducted in three independent environments, including the study of seedling resistance in greenhouse conditions, the study of APR at RIBSP (controlled inoculation), and APR at KRIAPI (uncontrolled inoculation). In both field conditions, phenotyping of APR to LR and SR was performed in two independent replicates at the stage of grain ripening with the maximum level of disease manifestation. Disease assessment was performed using the scale of Stakman for SR [33] and the scale of Mains and Jackson for LR [40]. The severity of rust infection on leaf and stem surfaces was evaluated using the modified Cobb scale [41,42]. To meet the data format required for linkage analysis, the results of LR and SR evaluations at both seedling and adult plant-growth stages were converted to the 0–9 linear disease scale as described by Zhang et al. [43].

To identify the influence of LR and SR severity on the productivity of the studied population, two important yield-related components, thousand kernel weight (TKW, g) and kernel yield per plot (YP, g/m$^2$), were also evaluated.

*2.4. Statistical Analysis of Phenotypic Data and QTL Mapping*

Phenotypic data processing, descriptive statistics, and one-tailed correlation tests were performed with SPSS Statistics v. 22 (SPSS Inc., Chicago, IL, USA).

The composite interval mapping (CIM) method with the Kosambi mapping function was used for the detection of QTLs by Windows QTL Cartographer v2.5 [44]. The threshold value for the logarithm of odds (LOD) score was calculated based on 1000 permutations and was 3.0 for all experiments with a walking step of 1 cM. QTLs were detected for each environment and replication separately (seedling resistance in GH, APR at KRIAPI, and APR at RIBSP). QTLs identified in individual environments and/or replications overlapping in 20 cM intervals and associated with the same trait were considered as identical [45]. Genetic maps with QTLs were drawn using MapChart v. 2.32 software [46]. For the markers with the same positions, only one single nucleotide polymorphism (SNP) maker was selected for the map.

All genes present within the interval of 500 kb to the left and 500 kb to the right (1 Mb in total) from the peak marker were identified using the Ensembl Plant database [47]. As a reference, the genome of *T. aestivum* RefSeq v1 was used. The exact position of the peak SNP in the genome was determined using a BLAST tool [48]. Proteins and RNA gene products were identified using the UniProt database [49] via cross-reference from Ensembl Plant.

## 3. Results

### 3.1. Phenotyping Variations of Seedling and Adult Plant Resistance in Mapping Population

The values of the resistance to target diseases in parents and 98 RILs are summarized in Table 3.

**Table 3.** Descriptive statistics for leaf rust (LR) and stem rust (SR) resistance at two plant-growth stages in the Pamyati Azieva × Paragon mapping population.

| Env. | Plant-Growth Stage | Race (Disease) | Parents IT [1] | | RILs IT | | | | | |
|---|---|---|---|---|---|---|---|---|---|---|
| | | | PA | Par | Mean | Range | R (%) | MR (%) | MS (%) | S (%) |
| GH | Seedling | TQTMQ (LR) | 6 (3) | 5 (3-) | 6.1 ± 0.9 | 5–8 | 0 | 0 | 89.8 | 10.2 |
| | | TQKHT (LR) | 8 (4-) | 6 (3) | 5.9 ± 1.0 | 4–9 | 0 | 3.1 | 91.8 | 5.1 |
| | | TRTHT (LR) | 6 (3) | 6 (3) | 5.9 ± 0.8 | 4–8 | 0 | 1.0 | 96.9 | 2.1 |
| | Seedling | TKRTF (SR) | 8 (4-) | 8 (4-) | 7.0 ± 1.4 | 5–9 | 0 | 0 | 45.9 | 54.1 |
| | | PKCTC (SR) | 5 (3-) | 8 (4-) | 5.4 ± 1.8 | 1–9 | 1.0 | 22.4 | 58.2 | 18.4 |
| | | RKRTF (SR) | 6 (3) | 5 (3-) | 6.5 ± 1.4 | 5–9 | 0 | 0 | 64.3 | 35.7 |
| RIBSP | Adult | LR | 8 (30S) | 6 (50MS) | 3.8 ± 1.4 | 0–9 | 38.8 | 11.2 | 29.6 | 20.4 |
| | | SR | 9 (80S) | 8 (40S) | 6.6 ± 2.9 | 1–9 | 1.0 | 23.5 | 18.4 | 57.1 |
| KRIAPI | | LR | 6 (40MS) | 6 (40MS) | 6.6 ± 1.7 | 2–9 | 0 | 9.2 | 64.3 | 26.5 |
| | | SR | 0 (0) | 1 (10R) | 1.8 ± 2.3 | 0–8 | 56.1 | 26.5 | 18.4 | 1.0 |

[1]—parents IT scores are given in 0–9 numeric scale, traditional IT scores are given in parentheses. Env., environment; GH, greenhouse conditions; PA, Pamyati Azieva; Par, Paragon; IT, infection type; R, percentage of resistant lines (0–1 on 9-point scale); MR, percentage of moderately resistant lines (2–4 on 9-point scale); MS, percentage of moderately susceptible lines (5–7 on 9-point scale); S, percentage of susceptible lines (8–9 on 9-point scale).

The average seedling resistance of RILs to LR races was between 5.9 and 6.1 points, corresponding to the moderately susceptible (MS) level. The major part of the population belonged to the MS group, with only several lines observed in the susceptible (S) group. Several lines were also in the resistant (R) group to the races TQKHT and TRTHT. Parental cultivars demonstrated an MS level of resistance to studied LR races, except for PA, which was susceptible to the race TQKHT. As for seedling resistance to SR, the average level in the RILs population was MS to all three SR races. However, unlike in the case of LR, the distribution of lines among resistance groups was different. Races TKRTF and RKRTF were divided between MS and S groups with a dominance of the S reaction to TKRTF and MS reaction to RKRTF. R and moderately resistant (MR) levels were detected only in the race PKCTC. Levels of resistance in parental cultivars were similar to races TKRTF (S) and RKRTF (MS), but PA demonstrated higher resistance to the race PKCTC than Par.

At the adult plant stage, the reactions of parents and RILs to LR and SR were significantly different between the studied environments. At RIBSP, the average reaction of RILs to LR was MR, with almost even distribution among all possible reactions observed in the population. At KRIAPI, the average level of resistance was MS with a dominance of MS and S reactions in the population. The parental cultivars demonstrated the same reaction to LR at KRIAPI, but at RIBSP, Par was more resistant than PA. For SR at the adult plant stage at RIBSP, the majority of RILs were in the S group, and the average level was MS. At KRIAPI, the largest part of the population was in the R group and the average level was MR. Parental cultivars also were in the S group at RIBSP and the R group at KRIAPI.

The analysis of variance showed that the resistance of RILs to LR at the seedling growth stage was significantly affected by the RIL genotype ($p < 0.01$) and the race of LR pathogen ($p < 0.05$), but not by genotype × race interaction (Table 4). For SR resistance, all factors had a significant influence ($p < 0.001$) on the resistance at the seedling stage (Table 4).

**Table 4.** ANOVA of plant genotype (Geno), pathogen race (Race), and plant genotype × pathogen race (Geno: Race) effects on seedling resistance to leaf rust (LR) and stem rust (SR) in the Pamyati Azieva × Paragon mapping population.

| Factor | df | LR | | | SR | | |
|---|---|---|---|---|---|---|---|
| | | MeanS | F | p | MeanS | F | p |
| Geno | 97 | 2.506 | 1.630 | 0.001 | 5.380 | 2.679 | $8.39 \times 10^{-11}$ |
| Race | 2 | 5.420 | 3.525 | 0.031 | 136.940 | 68.180 | $<2.00 \times 10^{-16}$ |
| Geno: Race | 194 | 1.396 | 0.908 | 0.766 | 4.540 | 2.262 | $1.20 \times 10^{-10}$ |
| Residuals | 294 | 1.537 | | | 2.010 | | |

df, degree of freedom; MeanS, mean square.

*3.2. Correlations among SR and LR Seedling and Adult Plant Resistance and Influence of APR on the Yield-Related Traits*

Significant positive correlations were found among the reactions to all three LR races at the seedling stage, as well as between APR to LR at KRIAPI and seedling resistance to LR races TQTMQ and TRTHT (Table 5). APR to LR at KRIAPI was also positively correlated with APR to SR at KRIAPI and seedling resistance to SR race TKRTF. For the other SR races, race PKCTC had a positive correlation with APR to SR at KRIAPI, and race RKRTF was negatively correlated with LR race TQTMQ. The only significant correlation of APR to SR at RIBSP was associated with LR race TRTHT.

**Table 5.** Correlations among race-specific seedling resistance and adult plant resistance (APR) to leaf rust (LR) and stem rust (SR) in the Pamyati Azieva × Paragon mapping population.

| | LR TQTMQ | LR TQKHT | LR TRTHT | APR LR RIBSP | APR LR KRIAPI | SR TKRTF | SR PKCTC | SR RKRTF | APR SR RIBSP |
|---|---|---|---|---|---|---|---|---|---|
| LR TQKHT | 0.193 * | | | | | | | | |
| LR TRTHT | 0.207 * | 0.283 ** | | | | | | | |
| APR LR RIBSP | 0.008 $^{ns}$ | 0.122 $^{ns}$ | 0.154 $^{ns}$ | | | | | | |
| APR LR KRIAPI | 0.251 ** | −0.028 $^{ns}$ | 0.192 * | −0.124 $^{ns}$ | | | | | |
| SR TKRTF | −0.152 $^{ns}$ | 0.000 $^{ns}$ | 0.141 $^{ns}$ | 0.088 $^{ns}$ | 0.228 * | | | | |
| SR PKCTC | −0.150 $^{ns}$ | 0.021 $^{ns}$ | 0.058 $^{ns}$ | 0.081 $^{ns}$ | −0.009 $^{ns}$ | −0.019 $^{ns}$ | | | |
| SR RKRTF | −0.202 * | −0.006 $^{ns}$ | 0.091 $^{ns}$ | −0.125 $^{ns}$ | 0.069 $^{ns}$ | 0.050 $^{ns}$ | 0.110 $^{ns}$ | | |
| APR SR RIBSP | −0.071 $^{ns}$ | 0.096 $^{ns}$ | 0.197 * | 0.130 $^{ns}$ | 0.038 $^{ns}$ | −0.075 $^{ns}$ | −0.040 $^{ns}$ | −0.029 $^{ns}$ | |
| APR SR KRIAPI | 0.127 $^{ns}$ | 0.040 $^{ns}$ | 0.117 $^{ns}$ | 0.018 $^{ns}$ | 0.245 ** | −0.006 $^{ns}$ | 0.182 * | 0.015 $^{ns}$ | −0.043 $^{ns}$ |

APR, adult plant resistance; $^{ns}$ not significant; * $p < 0.05$; ** $p < 0.01$.

The negative influence of LR and SR severities at the adult plant stage on the wheat YP and TKW was confirmed by significant negative correlations ($p < 0.01$) between these traits at RIBSP (Table 6). At KRIAPI, the severity of LR at the adult plant stage was negatively correlated with YP only.

**Table 6.** Correlations between leaf rust (LR) and stem rust (SR) resistance at the adult plant stage and yield-related traits in the Pamyati Azieva × Paragon mapping population.

| | TKW RIBSP | YP RIBSP | TKW KRIAPI | YP KRIAPI |
|---|---|---|---|---|
| LR RIBSP | −0.175 ** | −0.252 ** | – | – |
| SR RIBSP | −0.490 ** | −0.474 ** | – | – |
| LR KRIAPI | – | – | −0.055 $^{ns}$ | −0.200 * |
| SR KRIAPI | – | – | 0.134 $^{ns}$ | −0.151 $^{ns}$ |

TKW, thousand kernel weight (g); YP, yield per plot (g/m$^2$); $^{ns}$ not significant; * $p < 0.05$; ** $p < 0.01$.

### 3.3. Identification of QTLs for Seedling and Adult Plant Resistance to LR in the RIL Population

A total of 11 QTLs for resistance to LR were identified at the seedling and adult plant-growth stages. Out of these 11 QTLs, eight QTLs were detected for different LR races at the seedling stage, two QTLs were for APR, and one QTL was observed for both seedling and adult plant resistance (Table 7, Figure 1). QTLs for LR resistance were located on 10 chromosomes of the A, B, and D genomes. The phenotypic variations explained by an individual QTL ranged from 11.6% to 25.7%. Because all QTLs for LR resistance identified in this study explained more than 10% of the phenotypic variation, they were considered major QTLs [50]. The LOD score of QTLs for LR resistance was in the range of 3.2–8.6.

For the LR race TQTMQ, three QTLs identified on chromosomes 4A, 5B, and 7B were revealed. They explained 12.1–15.7% of the phenotypic variations. The alleles of all three QTLs associated with the increase in resistance to LR originated from PA. For the second LR race TQKHT, two QTLs on chromosomes 6A and 7D explained 25.7% and 11.6% of the variations in phenotype, respectively. Both QTLs associated with higher resistance to race TQKHT originated from Paragon. For the third LR race TRTHT, three QTLs on chromosomes 3A, 4D, and 6A were observed. Identified QTLs explained 12.3–16.2% of the variation in resistance to race TQKHT. The alleles of QTLs *QLr.ipbb-3A.2* and *QLr.ipbb-6A.5* increasing resistance were from Paragon, and the allele of *QLr.ipbb-4D.1* was from PA.

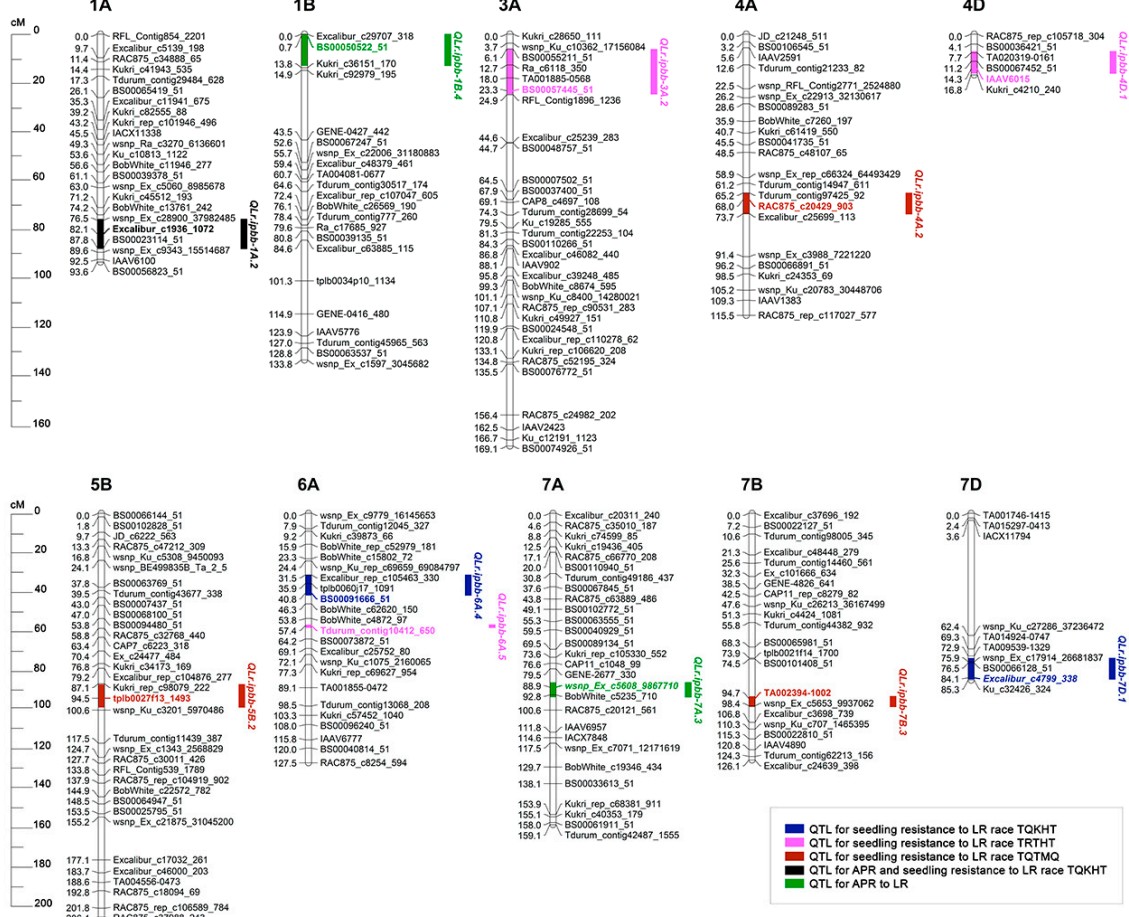

**Figure 1.** Pamyati Azieva × Paragon genetic map with quantitative trait loci (QTLs) for adult plant resistance (APR) to leaf rust (LR) in two regions and seedling resistance to three LR races. The region containing the QTL is indicated by a vertical bar on the right and followed by the name of the QTL. Single nucleotide polymorphism (SNP) markers are shown on the right and their genetic positions (cM) on the left. The peak marker for each QTL is highlighted in color and bolded. Colors of QTL indicate APR or race-specific seedling resistance.

**Table 7.** The list of quantitative trait loci (QTLs) for adult plant resistance (APR) and race-specific seedling resistance to leaf rust (LR) identified in the Pamyati Azieva × Paragon recombinant inbred lines (RILs) population.

| Race | Env. | QTL | Flanking Markers (Left–Right) | Chr. | CI (cM) | Peak (cM) | Max. LOD | $R^2$ (%) | Add. Effect [1] | Allele [1] |
|---|---|---|---|---|---|---|---|---|---|---|
| TQTMQ | GH | *QLr.ipbb-4A.2* | IAAV7104–Excalibur_c25699_113 | 4A | 65.2–73.9 | 68.0 | 3.3 | 12.1 | −0.47 | PA |
| | GH | *QLr.ipbb-5B.2* | Kukri_rep_c98079_222–BS00075815_51 | 5B | 87.4–99.2 | 94.5 | 4.3 | 15.2 | −0.57 | PA |
| | GH | *QLr.ipbb-7B.3* | BS00023023_51–wsnp_Ex_c5653_9937062 | 7B | 93.5–98.9 | 94.7 | 4.9 | 15.7 | −0.79 | PA |
| TQKHT | GH | *QLr.ipbb-6A.4* | Excalibur_rep_c105463_330–Ku_c37893_495 | 6A | 31.5–42.0 | 40.8 | 8.6 | 25.7 | 0.63 | Paragon |
| | GH | *QLr.ipbb-7D.1* | Kukri_c16416_647–BS00062644_51 | 7D | 74.0–84.9 | 84.1 | 4.2 | 11.6 | 0.41 | Paragon |
| TRTHT | GH | *QLr.ipbb-3A.2* | Excalibur_c74666_291–RFL_Contig1896_1236 | 3A | 6.2–24.7 | 23.3 | 5.1 | 16.2 | 0.46 | Paragon |
| | GH | *QLr.ipbb-4D.1* | TA020319-0161–BS00022436_51 | 4D | 7.1–16.2 | 14.3 | 3.7 | 12.3 | −0.39 | PA |
| | GH | *QLr.ipbb-6A.5* | BobWhite_c30930_192–BS00022992_51 | 6A | 56.9–58.5 | 57.4 | 4.8 | 15.1 | 0.44 | Paragon |
| APR | RIBSP | *QLr.ipbb-1B.4* | Excalibur_c29707_318–Kukri_c36151_170 | 1B | 0–13.0 | 0.7 | 3.2 | 13.6 | 1.29 | Paragon |
| APR | RIBSP | *QLr.ipbb-7A.3* | Ra_c4601_2417–CAP7_c7296_88 | 7A | 86.4–94.0 | 88.9 | 3.4 | 11.7 | 1.18 | Paragon |
| TQKHT/APR | GH/KRIAPI | *QLr.ipbb-1A.2* | RAC875_c12348_720–BS00022824_51 | 1A | 75.9–88.2 | 82.1 | 4.6 | 16.0 | 0.86 | Paragon |

CI, confidence interval; LOD, logarithm of odds; $R2$, phenotypic variance explained by the QTL; Chr., chromosome; Add., additive effect. [1]—Additive effect of QTL indicates increasing of the trait expression explained by the allele of one parent (positive for PA and negative for Paragon). In the case of disease resistance, increased expression of the trait is undesired, and the effective allele is taken from the other parent (negative for PA and positive for Paragon).

Two QTLs for APR to LR at RIBSP were identified on chromosomes 1B and 7A and explained 13.6% and 11.7% of phenotypic variation, respectively. For both APR QTLs, alleles associated with increased LR resistance were from Paragon. One QTL for APR to LR at KRIAPI was also detected at the seedling stage for the resistance to LR race TQKHT on chromosome 1A. It explained 16.0% of LR resistance variation. In both cases, alleles associated with higher resistance originated from Paragon.

### 3.4. QTLs for SR Resistance at Seedling and Adult Plant Stages Identified in PA × Par Mapping Population

A total of 13 QTLs were detected in this study for SR resistance at the seedling and adult plant-growth stages. Among them, seven race-specific QTLs were identified at the seedling stage (three QTLs for race TKRTF, three QTLs for race PKCTC, and one QTL for race RKRTF), three QTLs were observed for APR (two QTLs at KRIAPI and one QTL at RIBSP), and three QTLs were revealed in both the seedling and adult stages (Table 8, Figure 2). The identified QTLs for SR resistance were distributed among nine chromosomes of A, B, and D genomes and explained from 8.9% to 39.1% of the variation in the resistance to SR. In total, 11 out of 13 QTLs for SR resistance had $R^2 > 10\%$ and could be considered major QTLs. The LOD score for the detected QTL varied from 3.0 to 6.8.

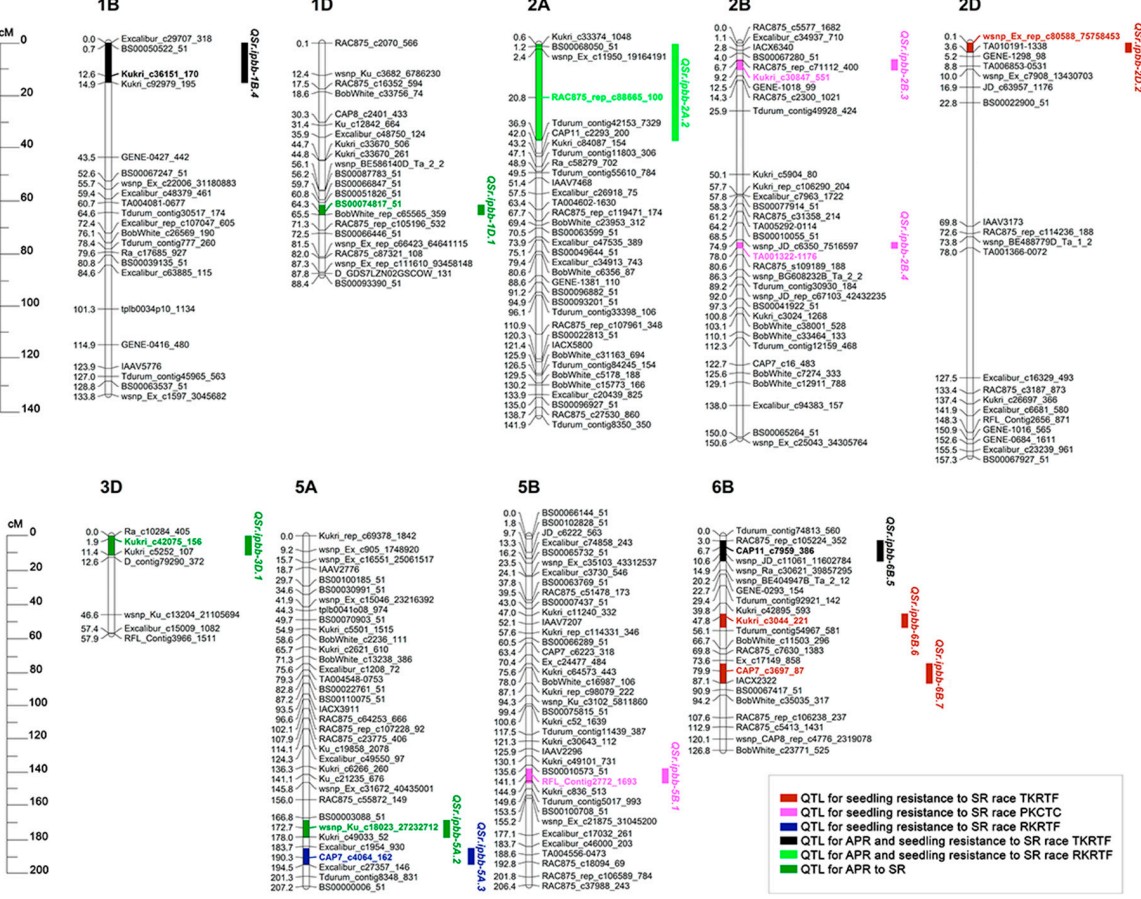

**Figure 2.** Pamyati Azieva × Paragon genetic map with quantitative trait loci (QTLs) for adult plant resistance (APR) to stem rust (SR) in two regions and seedling resistance to three SR races. The region containing the QTL is indicated by a vertical bar on the right and followed by the name of the QTL. Single nucleotide polymorphism (SNP) markers are positioned on the right and their genetic positions (cM) are shown on the left. The peak marker for each QTL is highlighted in color and bold. Colors of QTL indicate APR or race-specific seedling resistance.

**Table 8.** The list of quantitative trait loci (QTLs) for APR and race-specific seedling resistance to stem rust (SR) identified in Pamyati Azieva × Paragon recombinant inbred lines (RILs) population.

| Race | Env. | QTL | Flanking Markers (Left–Right) | Chr. | CI (cM) | Peak (cM) | Max. LOD | R2 (%) | Add. Effect * | Allele * |
|---|---|---|---|---|---|---|---|---|---|---|
| TKRTF | GH | *QSr.ipbb-2D.2* | wsnp_Ex_rep_c80588_75758453–TA010191-1338 | 2D | 0–3.7 | 0.1 | 3.0 | 10.0 | 0.43 | Paragon |
| | GH | *QSr.ipbb-6B.6* | wsnp_Ex_c12618_20079758–wsnp_Ex_c1249_2399894 | 6B | 46.0–54.3 | 47.8 | 4.2 | 14.1 | −0.57 | PA |
| | GH | *QSr.ipbb-6B.7* | wsnp_Ku_c43368_50890819–wsnp_Ku_c4910_8793327 | 6B | 75.5–87.3 | 79.9 | 4.7 | 19.2 | −0.66 | PA |
| PKCTC | GH | *QSr.ipbb-2B.3* | RAC875_rep_c71112_400–Kukri_c1175_1577 | 2B | 6.3–10.4 | 9.2 | 3.9 | 12.2 | −0.63 | PA |
| | GH | *QSr.ipbb-2B.4* | BS00041323_51–RAC875_c32503_134 | 2B | 75.9–78.3 | 78.0 | 2.9 | 8.9 | 0.54 | Paragon |
| | GH | *QSr.ipbb-5B.1* | RAC875_rep_c114200_428–wsnp_RFL_Contig1548_762547 | 5B | 137.6–146.3 | 141.1 | 4.9 | 14.7 | 0.96 | Paragon |
| RKRTF | GH | *QSr.ipbb-5A.3* | BS00036851_51–Excalibur_c27357_146 | 5A | 184.7–194.4 | 190.3 | 6.8 | 24.7 | 0.72 | Paragon |
| APR | KRIAPI | *QSr.ipbb-1D.1* | BS00051826_51–BobWhite_rep_c65565_359 | 1D | 61.7–65.5 | 64.3 | 5.0 | 18.0 | 2.24 | Paragon |
| | RIBSP | *QSr.ipbb-3D.1* | Ra_c10284_405–Kukri_c5252_107 | 3D | 0–11.4 | 1.9 | 3.1 | 11.3 | −0.99 | PA |
| | KRIAPI | *QSr.ipbb-5A.2* | BS00089076_51–CAP11_c2623_196 | 5A | 168.2–178.5 | 172.7 | 3.0 | 10.8 | 0.72 | Paragon |
| TKRTF/APR | GH/KRIAPI | *QSr.ipbb-1B.4* | Excalibur_c29707_318–Kukri_c92979_195 | 1B | 0–15.2 | 12.6 | 4.2 | 15.7 | 0.59/−0.77 | Paragon/PA |
| RKRTF/APR | GH/RIBSP | *QSr.ipbb-2A.2* | Kukri_c33374_1048–Tdurum_contig42153_5854 | 2A | 0.5–37.2 | 30.8 | 5.2 | 39.1 | −1.45 | PA |
| TKRTF/APR | GH/RIBSP | *QSr.ipbb-6B.5* | RAC875_rep_c105224_352–Kukri_rep_c107077_360 | 6B | 2.7–15.1 | 6.7 | 4.6 | 17.3 | −0.56/1.26 | PA/Paragon |

* Additive effect of QTL indicates increasing of the trait expression explained by the allele of one parent (positive for PA and negative for Paragon). In the case of disease resistance, increased expression of the trait is undesired, and the effective allele is taken from the other parent (negative for PA and positive for Paragon).

Three QTLs for resistance to SR race TKRTF were identified, including two QTLs mapped on chromosome 6B and one QTL on 2D. These QTLs explained 10.0–19.2% of the variation in SR resistance to race TKRTF. QTLs *QSr.ipbb-6B.6* and *QSr.ipbb-6B.7* had alleles increasing SR resistance to race TKRTF carried by PA, whereas resistance allele *QSr.ipbb-2D.2* originated from Paragon. For the second SR race PKCTC, two QTLs for resistance to this race were located on chromosome 2B and one QTL was on 5B. The phenotypic variance conditioned by these QTLs varied from 8.9% to 14.7%. The third SR race RKRTF allowed the identification of one race-specific QTL on chromosome 5A explaining 24.7% of the phenotypic variation. Its allele, associated with an increase in SR resistance, originated from Paragon.

Three QTLs for APR to SR were identified on chromosomes 1D, 3D, and 5A, and explained from 10.8% to 18.0% of SR resistance variation. Two QTLs identified at KRIAPI had alleles increasing resistance to SR originating from Paragon, and the allele of QTL at RIBSP was from PA. The last three QTLs for LR resistance occurred multiple times in the experiment and are located on chromosomes 1B, 2A, and 6B. The QTL *QSr.ipbb-1B.4* was detected as race-specific to TKRTF at the seedling stage and as APR QTL at KRIAPI. It explained 15.7% of SR resistance variation and had alleles increasing resistance originating from Paragon at the seedling stage and PA at the adult plant stage. The QTL *QSr.ipbb-2A.2* was identified as effective against SR race RKRTF at the seedling stage and as APR QTL at RIBSP. This QTL explained 39.1% of the phenotypic variation, and alleles increasing SR resistance at both growth stages were inherited from PA. The QTL *QSr.ipbb-6B.5* was discovered at the seedling stage to race TKRTF and at the adult plant stage at RIBSP. The QTL explained 17.3% of SR resistance variations. Its alleles increasing resistance originated from PA in the case of seedling resistance and from Paragon at the adult growth stage.

### 3.5. Comparison of Identified QTLs with Previous Works and Gene Identification

The QTLs identified in this study were analyzed in comparison with previously reported QTLs for LR and SR resistance in the PA × Par RILs population [27] and with QTLs for LR and SR resistance at RIBSP identified using genome-wide association study (GWAS) [51]. The location of each identified QTL was compared to the genetic positions of known *Lr* and *Sr* genes (Table 9). In total, four candidate *Lr* genes and four QTLs were found for five QTLs associated with LR resistance in this study. In the analysis of QTLs for SR resistance, we found similarities with the genetic locations of eight previously identified QTLs and/or candidate *Sr* genes.

**Table 9.** Comparison of quantitative trait loci (QTLs) for leaf rust (LR) and stem rust (SR) resistance identified in this study in a Pamyati Azieva × Paragon mapping population with previously described QTLs and candidate *Lr* and *Sr* genes.

| # | Trait | Type of Resistance (Race) | QTL | Reference QTL | Candidate Genes |
|---|---|---|---|---|---|
| 1 | | TQKHT/APR | *QLr.ipbb-1A.2* | *QLr.ipbb-1A.1* [51] | - |
| 2 | | APR | *QLr.ipbb-1B.4* | - | *Lr21* [47] |
| 3 | LR | TRTHT | *QLr.ipbb-3A.2* | - | *Lr63, Lr66* [52] |
| 4 | | TQKHT | *QLr.ipbb-6A.4* | *QLr.ipbb-6A.1* [51] | - |
| 5 | | TRTHT | *QLr.ipbb-6A.5* | *QLr.ipbb-6A.2* [51] | - |
| 6 | | TQTMQ | *QLr.ipbb-7B.3* | *QLr.ipbb-7B.1* [51] | *Lr14* [52] |
| 1 | | TKRTF/APR | *QSr.ipbb-1B.4* | - | *Sr31* [53] |
| 2 | | APR | *QSr.ipbb-1D.1* | - | *Sr18* [54] |
| 3 | | RKRTF/APR | *QSr.ipbb-2A.2* | - | *Sr32-2A* [53] |
| 4 | | PKCTC | *QSr.ipbb-2B.3* | - | *Sr32-2B, Sr39* [53] |
| 5 | SR | PKCTC | *QSr.ipbb-2B.4* | *QSR.IPBB-2B* [26] | *Sr36* [53] |
| 6 | | TKRTF | *QSr.ipbb-2D.2* | - | *Sr32-2D* [53] |
| 7 | | TKRTF/APR | *QSr.ipbb-6B.5* | *QSR.IPBB-6B.1* [27], *QSr.ipbb-6B.3* [51] | - |
| 8 | | TKRTF | *QSr.ipbb-6B.7* | *QSR.IPBB-6B.2* [27], *QSr.ipbb-6B.4* [51] | *Sr11* [54] |

The region of each QTL was analyzed for the presence of protein-coding genes in the interval 500 kb upstream and 500 kb downstream from the most significant SNP (Table S1). The analysis of LR-associated QTL regions suggested the presence of 158 genes ranging from 6 (*QLr.ipbb-7B.3* and *QLr.ipbb-7D.1*) to 22 (*QLr.ipbb-1B.4* and *QLr.ipbb-6A.4*) genes per interval. A similar search for SR-associated QTL regions indicated the presence of 226 genes ranging from 5 (*QSr.ipbb-2B.4*) to 29 (*QSr.ipbb-1B.4*) genes per interval. Among these 158 genes identified for QTLs associated with LR, 48.9% coded for proteins with functions known in *T. aestivum*, 48.6% for uncharacterized proteins, and 2.5% for RNAs. For QTLs associated with SR, 56.6% of genes coded for proteins uncharacterized in *T. aestivum*, 41.2% described protein-coding genes, and 2.2% coded for RNAs. Among genes coding for uncharacterized proteins, sequences similar to the 24 QTL regions for LR and 39 QTL regions for SR were identified in other grass species (Table S1). Orthologous genes with their sequence similarity level higher than 70% were selected and are listed.

## 4. Discussion

### 4.1. General Resistance of RILs in Studied Environments

At the seedling stage, the majority of RILs and parental cultivars showed MS and S levels of resistance to all races of LR and SR, except for the SR race PKCTC, where several lines were identified as R and MR (Table 3). The ANOVA test showed a more significant influence of pathogen genotype (race) on the resistance of RILs rather than the genotype of wheat lines (Table 4). This result indicated that these genetic factors associated with resistance are race-specific. In the world and in Kazakhstan, breeding programs are mostly focused on the combination of seedling resistance and APR in new cultivars. Pyramiding of seedling gene(s) with slow rusting APR gene(s) usually results in higher resistance of the crop. This agrees with wheat R genes conferring resistance to LR (*Lr1*, *Lr10*, *Lr21*) and SR (*Sr22*, *Sr33*, *Sr35*, *Sr45*, *Sr50*) being cloned and widely used in wheat breeding [55]. However, the significant positive correlations among LR races observed in this study (Table 5) suggested the involvement of genetic factors that are effective against all three races. The presence of strong positive correlations between APR to LR and SR at KRIAPI also indicated that genes conferring LR resistance are either closely linked or may have a pleiotropic effect on genes that control SR resistance [26,56]. Positive correlations were simultaneously observed between seedling resistance to LR races TQTHQ and TKTHT and APR to LR at KRIAPI, as well as between seedling resistance to SR race PKCTC and APR to SR at KRIAPI (Table 5). The relationship between race-specific seedling and broad adult plant resistances could be influenced by the presence of LR and SR races in the fields at KRIAPI. This also suggested that the wheat germplasm growing in this region could be effectively and rapidly screened for resistance to LR and SR at the seedling stage in a greenhouse [57].

LR and SR resistances are complex traits [58]; this was confirmed by the range of reactions to pathogens and the presence of transgressive segregations. Even when parents demonstrated the same level of resistance, such as APR to SR, RILs still showed transgressive phenotypes in the direction of either resistance (RIBSP) or susceptibility (KRIAPI) (Table 3). This phenomenon is not rare; it was previously described for many other quantitatively inherited wheat traits; for example, in studies of grain quality traits [22], grain Zn and Fe concentrations [59], grain yield and plant height [60], and rust diseases [61,62].

### 4.2. QTL Mapping for Leaf Rust Resistance

Alleles conferring increased resistance of QTLs for LR race TQKHT and APR at RIBSP originated from Par (Table 7). The higher LR resistance of Par in comparison with PA indicated that the U.K. cultivar is a promising source for wheat breeding programs in Kazakhstan. PA was simultaneously found to be a source for QTLs with increased LR resistance to race TQTMQ.

The 11 QTLs for the resistance to LR at the seedling and adult plant-growth stages can be divided into two categories: (1) similar to QTLs previously detected for LR resistance and (2) presumably

novel QTLs. The first category consisted of 6 out of 11 QTLs for LR resistance (Table 9). Four of the QTLs for LR resistance with similar genetic positions (*QLr.ipbb-1A.2* (APR at KRIAPI), *QLr.ipbb-6A.4* (seedling resistance to TQKHT), *QLr.ipbb-6A.5* (seedling resistance to TRTHT), and *QLr.ipbb-7B.3* (seedling resistance to TQTMQ) were previously identified in a GWAS study performed at RIBSP in 2018/2019 at the adult plant stage [50]. Hence, multiple occurrences of QTLs associated with the resistance to LR in different conditions and environments indicated the broad stability of these loci. *QLr.ipbb-7B.3* may be associated with the gene *Lr14* located in a similar region of the genome (Table 9). The effectiveness of allele *Lr14a* was described for northern Kazakhstan and *Lr14b* for eastern and western Kazakhstan [7]. *Lr14* was also described as an effective resistance factor to TQTMQ (Table 1). The APR QTL *QLr.ipbb-1B.4* is associated with the gene *Lr21*, positioned in close proximity to the peak of the QTL (Table S1). This gene was described as effective in southeastern Kazakhstan [7]. The last QTL from the first group, *QLr.ipbb-3A.2*, is probably associated with genes *Lr63* and *Lr66* (Table 9). Unfortunately, information is lacking about the role of these genes in the wheat-growing areas of Kazakhstan. However, *Lr63* and *Lr66* are known to condition low to intermediate infection types to most of *P. recondita* isolates [63]. The remaining five QTLs identified for LR resistance are presumably novel genetic factors, since there were no reliable matches between their positions in the genome and previously identified QTLs or genes.

### 4.3. QTLs for Stem Rust

In 13 QTLs for the resistance to SR identified in this study, alleles presumably increasing resistance originated from both PA and Par (Table 8). Similar to LR resistance, SR-resistance-associated QTLs could be divided into two loci groups, where the first group has similar genetic positions with previously reported QTLs for SR resistance (Table 9), and the second group has none of those matches. The first group includes 8 out of 13 QTLs identified for SR resistance. For three of them—*QSr.ipbb-2B.4* (seedling resistance to PKCTC), *QSr.ipbb-6B.5* (seedling resistance to TKRTF and APR in RIBSP), and *QSr.ipbb-6B.7* (seedling resistance to TKRTF)—QTLs for SR resistance with similar positions in the genome were identified in a previous work involving the PA × Par mapping population [27] and in a GWAS study using resistance data obtained from RIBSP [51]. Similar to the LR study, these findings may indicate the stability of identified QTLs. In addition to the information with QTL similarities, several specific *Sr* genes seem to be associated with QTLs from this study (Table 9). One of the most interesting findings was the identification of three QTLs on distal ends of chromosomes 2A, 2B, and 2D responsible for seedling resistance to SR races RKRTF, PKCTC, and TKRTF, respectively. These QTLs could be associated with the gene *Sr32*, which was mapped in these regions of chromosomes 2A [64], 2B [65], and 2D [66]. The gene was previously reported as effective against Ug99 and related SR races [66]. The other *Sr* genes involved in resistance to SR races in the Ug99 lineage and possibly associated with QTL from this study are *Sr31* (resistant to TTKSF and TTKSP), *Sr36* (all Ug99 lineage races, except TTTSK), and *Sr39* (all Ug99 lineage races) [67]. Among the SR races used in this study, *Sr36* was described as effective against PKCTC (Table 1). The resistance pattern is similar to *QSr.ipbb-2B.4*, which is located in a nearby region of the chromosome. The second group of the genetic factors consisted of the remaining five QTLs that could be novel QTLs associated with resistance to SR.

### 4.4. QTLs Cluster on Chromosome 1B

The QTLs associated with several traits are common in wheat. It may occur due to pleiotropic effect or their tight linkage. For the resistance to wheat fungal diseases, pleiotropic APR genes *Lr34/Yr18/Pm38/Sr57* [68], *Lr46/Yr29/Pm39/Sr58* [69], and *Lr67/Yr46/Pm46/Sr55* [70] were previously described. Among the QTLs identified for LR and SR resistance in this study, two QTLs (*QLr.ipbb-1B.4* and *QSr.ipbb-1B.4*) occupy the same interval on chromosome 1B (Tables 7 and 8, Figures 1 and 2). In addition, the *QLr.ipbb-1B.4* interval contains the resistance gene *Lr21* less than 500 kb from the significant peak, whereas the interval of *QSr.ipbb-1B.4* has genes for disease resistance proteins and resistance-related kinases next to the peak marker (Table S1). *Lr21* was described as effective

for southeastern Kazakhstan [7]. Common markers in these intervals suggest the usefulness for marker-assisted breeding of these QTLs to develop wheat cultivars with durable rust resistance for gene pyramiding [11].

## 5. Conclusions

Overall, 24 QTLs for the resistance to rust diseases at the seedling and adult plant stages were identified in this study, including 11 QTLs for LR and 13 QTLs for SR. Among the QTLs associated with LR, eight QTLs were race-specific and detected at the seedling stage, two QTLs were at the stage of the adult plant, and one QTL was identified in both stages. The QTLs for LR-resistance explained from 11.6% (*QLr.ipbb-7D.1*) to 25.7% (*QLr.ipbb-6A.4*) of the phenotypic variation and were detected on 10 chromosomes. The increased resistance to LR in TQTMQ race-specific QTLs originated from PA; in QTLs specific for the race TQKHT and APR, alleles were from Par. For TRTHT, the origin of resistance alleles in identified QTLs was both parental cultivars. For SR resistance, seven QTLs were race-specific and detected at the seedling stage, three QTLs were identified at the adult plant stage, and three QTLs were identified at both growth stages. SR-associated QTLs explained from 8.9% (*QSr.ipbb-2B.4*) to 39.1% (*QSr.ipbb-2A.2*) of variation in SR resistance and were mapped on nine chromosomes. The alleles increasing resistance to SR originated from both parents: effective alleles in six QTLs were from Par, in five QTLs from PA, and two QTLs had a different origin of resistance at the seedling and adult plant stages. Among the QTLs from this study, 10 QTLs were putative and 14 matching QTLs were found in previous works involving the PA × Par population, a GWAS study at RIBSP, and possible candidate resistance genes. The cluster of QTLs associated with both LR and SR resistances was identified on chromosome 1B. Thus, the QTLs revealed in this study may play an essential role in the improvement of wheat resistance to LR and SR via marker-assisted selection.

**Supplementary Materials:** The following are available online at http://www.mdpi.com/2073-4395/10/9/1285/s1, Table S1. The list of protein- and RNA-coding genes 500 kb upstream and 500 kb downstream from the most significant SNP of the QTL.

**Author Contributions:** Conceptualization, S.A. and Y.T.; methodology, A.R. and Y.T.; formal analysis, Y.G. and G.Y.; investigation, Y.G., A.R., and S.A.; resources, Y.T. and A.R.; data curation, Y.G., G.Y., and Y.T.; writing—original draft preparation, Y.G.; writing—review and editing, Y.G., S.A., A.R., G.A., and Y.T.; supervision, Y.T.; project administration, S.A.; funding acquisition, A.R. All authors have read and agreed to the published version of the manuscript.

**Funding:** This research was funded by the Ministry of Agriculture of the Republic of Kazakhstan, grant number BR06249329. The APC was also funded by grant number BR06249329.

**Acknowledgments:** This work was conducted within the framework of the project "Development of new DNA markers associated with the resistance of bread wheat to the most dangerous fungal diseases in Kazakhstan" in Program "Development of the innovative systems for increasing the resistance of wheat varieties to especially dangerous diseases in the Republic of Kazakhstan": BR06249329 supported by the Ministry of Agriculture of the Republic of Kazakhstan.

**Conflicts of Interest:** The authors declare no conflict of interest.

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
