# Peer review of "QTL Mapping for Seedling and Adult Plant Resistance to Leaf and Stem Rusts in Pamyati Azieva × Paragon Mapping Population of Bread Wheat"

_agronomy, doi:10.3390/agronomy10091285_

Round 1
Reviewer 1 Report
Dear Authors,
Please, some more comments are attached. Please, double-check some of the previous comments. It would be great if the manuscript properly improved. Please, include seedling and APR IT in Tables 7 and 8.
I look forward to a new improved version.

Author Response
Dear Reviewer 1.
The authors appreciate you very much for all your efforts to improve the manuscript. Below we provided replies to all your comments in the second round of reading of the manuscript. Again, thank you very much for your contribution to the improvement of the manuscript.
Sincerely Yours,
Yerlan Turuspekov
Commented [Rev1]: It seems, you are converted both seedling and adult to this scale, but please describe this.
Thank you for the remark. We corrected this sentence. Lines 139-141.
Commented [Rev2]: I think you can move up this statement for this subchapter.
Thank you for the suggestion. This part is moved to subchapter 2.1. “Plant Material and Genotyping”. Lines 99-102.
Commented [Rev3]: The Table still is comples, and please simplify this Table. As you indicated, the IT6 in APR has 40MS, 50MS, and 60MS or IT8 has 10S, 20S, 30S and 40S, etc. So, in your study, in which IT are you referring? For example, Par IT6 correspond to which reaction? Possibly, 30MS may have 2/3 QTLs, 40MS 1/2 QTLs, etc.
Thank you for the note. We added the traditional IT score of both resistances types in parentheses to Parents IT column together with a linear scale. In regards to your last sentence related to QTL, the Table describes the general resistance of the mapping population to different races at seedling and adult stages and revealing general variability in total population, and, therefore, identified QTLs could not be linked to this Table. QTL were identified by using MP lines that have contrasting genetic profiles for these loci and differences in the phenotype, while this Table reflects resistance in the total population. The effect of each QTL on the resistance was given in Tables 7 and 8.
Commented [Rev4]: Since you are using different races in the APR compare to seedling, then you need to describe in the results part. In addition, the APR reactions may have been inducing by beneficial soil microbiome. Many QTLs are tightly linked with the beneficial soil microbiome, which it will be great to consider in the future. It would be good if you could highlight in a discussion about the role of the beneficial soil microbiome in the field.
Thank you for your comment. Unfortunately, we did not identify the race composition at the adult plant stage, as well as the soil microbiome in two regions. Therefore, we cannot be sure that different reaction of the MP to pathogens between RIBSP and KRIAPI was influenced by different races, weather conditions, or soil microbiome. We agree that it will be interesting to compare the composition of the beneficial soil microbiome in these two environments and their effect on the resistance in our future studies.
Commented [Rev5]: Please, indicate due to different races.
Thank you for the suggestion, but this is Result section, and as it was mentioned above, we cannot be sure that different reaction of the MP to pathogens between RIBSP and KRIAPI was influenced by different races.
Commented [Rev6]: Please, include seedling and APR IT in this Table, to get an overview QTLs about effects. However, when you add IT6, etc, please make sure to add disease reaction.
Thank you for the suggestion. We can judge the effect of each individual QTL on plant resistance by the Additive effect and R2 of this QTL. The additive effect shows how the allele of the gene in one of the parents is generally increasing resistance in MP. It has no correlation with IT in the total population. R2 shows the percentage of total resistance, presumably explained by this particular QTL. We think the addition of IT in this Table will only give misleading interpretations about QTLs’ effect.
Commented [Rev7]: Please, do the same, include seedling and APR IT in this Table, to get an overview of QTLs effects. However, when you add IT6, etc, please make sure to add disease reaction.
Thank you for your comment. The answer will be similar as in reply to your previous comment.

Reviewer 2 Report
Thanks for taking my suggestions on board and for the additional explanations provided. I went through your edits and tweaks and noticed only a few ultra-minor typos, as detailed below:
Line 365: “rusdt” should be “rust”.
Line 398: “PA x Par” has been permutated (i.e. it appears twice in the sentence).
Line 479: “crops” should be “crop”.
Line 556: “on the chromosome 1B” should be “on chromosome 1B”
Lines 87 to 90: I suggest a few minor tweaks to make it read:
“Hence, the current study adds the investigation of seedling resistance in the MP to LR and SR races. In addition to one-year studies in South-east and North Kazakhstan, this work covers the analysis of LR and SR resistance in the MP in South Kazakhstan in 2018 and 2019.”
Author Response
Replies to Reviewer 2.
Dear colleagues, we deeply appreciate you for all your help in the improvement of the manuscript. We accepted all suggestions in your round 2 reading.
Best wishes,
Yerlan Turuspekov
Line 365: “rusdt” should be “rust”.
Thanks for the comment. We corrected the typo.
Line 398: “PA x Par” has been permutated (i.e. it appears twice in the sentence).
Thanks for the comment. We corrected the sentence.
Line 479: “crops” should be “crop”.
Thanks for the comment. We corrected the typo.
Line 556: “on the chromosome 1B” should be “on chromosome 1B”
Thanks for the comment. The sentence corrected.
Lines 87 to 90: I suggest a few minor tweaks to make it read:
“Hence, the current study adds the investigation of seedling resistance in the MP to LR and SR races. In addition to one-year studies in South-east and North Kazakhstan, this work covers the analysis of LR and SR resistance in the MP in South Kazakhstan in 2018 and 2019.”
Thank you for the suggestion. We revised the sentence.

This manuscript is a resubmission of an earlier submission. The following is a list of the peer review reports and author responses from that submission.
Round 1
Reviewer 1 Report
Title: QTL Mapping for Seedling and Adult Plant Resistance to Leaf and Stem Rusts in Pamyati Azieva Paragon Mapping Population of Bread Wheat
Authors: Yuliya Genievskaya, Saule Abugalieva, Aralbek Rsaliyev, Gulbahar Yskakova, and Yerlan Turuspekov
The present manuscript described the QTL mapping to stem rust and leaf rust in RILs.
At present, novel devastating Sr and Lr races are emerging and invading wheat production, which mostly leads to mass fungicides applications. The use of genetic resistance to rusts pathogens is the most economic great value to wheat growers and society to avoid fungicide application, environmental pollution, and social reasons. Thus, the authors try to find R genes for breeding programs in Kazakhstan.
This paper requires substantial review for publication in its present form. Despite inconsistencies, the material could be published but requires serious input from all authors to ensure that interpretations are correct.
Interpretation of data of the type presented is extremely difficult for the average reader and invariably results in QTLs identification. Only relatively few specialists can reasonably sort and critically analyze the material as presented.
In the following paragraphs, I will try to outline some issues, which the authors have to clarify, and if necessary, change, delete or edit.
Materials and Methods
Lines 80 to 88: The goal needs to be described more clearly
Lines 85 to 87: Are the Lr and Sr race analysis one of the goals of this manuscript?
Lines 99 to 134: Please, describe details of experimental design in seedling and adult stages. It would be good to describe, separate for seedling and adult testing. How have you used spore, fresh, or retrieved from the freezer? Describe more precisely and details on how conducted the seedling and APR. Any replications for these experiments in both field locations.
Line 122: In the RIBSP mix of Lr and Sr races were inoculated in the field. However, it’s not mentioned which races. Which races are applied?
Line 123: In KIRAPI, natural infection of Lr and Sr has occurred. Have you done any race analysis to know which races have been existed in the field?
Lines 149 to 151: A bit detailed explanation needed for alignment against T. aestivum RefSeq v1. However, in the result part, this alignment against T. aestivum RefSeq v1 is not clearly stated.
The genotyping method is missing in the material and method part. Which genotyping platform have you used? How have you isolated DNA for genotyping? How have you done bioinformatics analysis, etc.?
Results
Table 3:
It seems seedling infection types (IT) are the same for Lr, except race TQKHT. Even though, it’s a bit difficult to indicate if its seedling resistance with the indicated IT. Exactly, the same scenario for the Sr seedling test.
Parents (PA and Par): Is it infection types? It would be good to indicate this
Range: I guess this is also ITs, and it’s also good to indicate in the table.
Why in seedling stage 3.1% MR in TQKHT, 1% MR in TRTHT, and 1% R in PKCTC races, or some cross-pollination have occurred in the greenhouse in the early generation?
If possible, please mention the name of mixed races in both locations for APR to Lr and Sr.
How did you find QTL in the seedling stage? This is not clearly described.
Lines 159 to 161: The major part of the population belonged to the MS group,……. This sentence is not clearly stated. Please rewrite for a better explanation. How some RILs show resistance to TQKHT and TRTHT. What can be an explanation for this resistant reaction?
Lines 165 to 168: You have mentioned the reaction to Sr has been divided into MS and S reaction to all three races. How the MR and R reactions were identified for the PKCTC race?
Lines 169 to 171: At RIBSP the parental shows high infection type MS to S reactions. Like seedling reactions, how are they become resistant at the adult stage to Lr?
Lines 171 to 172: The same in KIRAPI. How it turned out to be 9.2% MR to Lr?
Lines 173 to 174: Please, check the parental reactions in both locations to Lr. It seems not so many differences for Lr in both locations.
Lines 174 to 177: Both parental highly susceptible in RIBS and MS in KIRAPI to Sr. How the RILs turned out to be resistant and moderately resistant in both location to Sr?
It’s a bit difficult for QTL mapping for both rusts, in this study. Because, the seedling infection types are the same, therefore it’s hard for QTL mapping.
Lines 178 to 181: If you correct the above suggestion, these need to be changed along with Table 4.
Lines 153 to 181: It would be good to have a separate subheadings for Lr and Sr. Now its mix of results for both rusts, which the reader will not be easy to follow the results you are presented in this manuscript.
Lines 186 to 196: How are they significantly correlating in seedling and adult stages? Please, check the above comments.
Lines 199 to 268: Detecting QTLs for Seedling and APR to Lr and Sr:
How did you find QTLs for the seedling adult plant stages? Because the infection types are more-less similar in parental lines. Please, check the above suggestions for details.
Still need to be clearly described, how the RILs turned out to be R and MR in both stages to Lr and Sr.
It seems the seedling and adult plant testing is not strongly supporting for identifying QTLs.
The putative SNP can be converted to KASP or other markers, and then validate the intending QTL
If the results will be changed according to the above suggestions, then it will lead to a different discussion. Then, please, change the discussion part accordingly.
This manuscript requires substantial revisions
Reviewer 2 Report
Genievskaya and colleagues tested a wheat RIL population in two field locations in Kazakhstan and under greenhouse conditions for seedling and adult plant resistance to leaf rust and stem rust. The authors identified 24 resistance QTLs. They then proceed to describe these QTLs in terms of their effect size, race specificity, seedling vs APR, candidate genes in the region, and co-localisation with known Lr and Sr genes.
The study appears to be sound and the main statements and conclusions are supported by the results.
I do have one important concern, however, which is that it would appear that the authors have twice previously phenotyped the very same RIL population for rusts in Kazakhstan (Genievskaya et al. 2019; references 26 and 27). It light of this, it was not clear to me how the present study differentiates itself from these two previous studies, nor what the motivation was for the present study. This should be made clear in the introduction.
Apart from this, I only have some minor concerns as detailed below:
- Lines 66-67. The authors mention a series of APR Lr genes but omit to mention the two most famous ones, namely the cloned Lr34 and Lr67 genes. This seems like a strange omission.
- In the methods section, the study in which the RIL population was generated and genotyped is referred to. It would be good to provide a short summary of its salient features. As it is, we are only told that the size of the population is 98 RILs. However, what was the genotyping platform that was used? How many markers made it onto the genetic map? What is the marker density? This information would help with the reader’s interpretation of the results.
- Lines 297-298. It is mentioned that the majority of wheat breeding programs focus on APR genes. Is this also the case in Kazakhstan, in particular in the breeding program that gave rise to the RIL parent Pamiyati Azieva?
- Lines 370 to 371. “share the same chromosome region on 1B” – as what? This is not implicit from the grammar.
- I spotted some minor typos and grammatical errors, as follows:
- Line 39. Start sentence with: “The primary goals…”
- Line 42. It should be “the Puccinia genus”
- Line 43. “Tritici” should be “tritici”.
- Line 46. I suggest replacing “wheat field” with “wheat crop”.
- Line 57. “groups” should be “group”.
- Line 95. Should “originated” not be “originating”?
- Line 102. “growing” should be “growth”
- Line 183. This should read: “… in the RIL population” or “in the RILs”.
- Table 4. I don’t think Geno is meant to be in bold.
- Line 191 should SR not be LR here?
- Line 199. This should be: “… in the RIL Population”
- Line 237. Table 7. Title should read: “… in the PA x PAR RIL population.”
- Line 195. Should be “via marker-assisted selection.”
- Table 7. Add. Effect and Allele both carry the same superscript. Is this intentional? In the bottom of the Table, the superscript 1 is followed by an underscore (i.e. “_”). Is this intentional?
- Lines 366-367. The grammar in this sentence needs fixing.
- Lines 370-371: Should be: “…share the same genome region on chromosome 1B…”